# Uncertainty in the evolution of northwest North Atlantic circulation leads to diverging biogeochemical projections

Krysten Rutherford[1,2], Katja Fennel[1], Lina Garcia Suarez[1], Jasmin G. John[3,4]

[1]Department of Oceanography, Dalhousie University, Halifax, NS, B3H 4R2, Canada

[2]Institute of Ocean Science, Fisheries and Oceans Canada, Sidney, BC, V8L 4B2, Canada

[3]NOAA/OAR/Atlantic Oceanographic and Meteorological Laboratory, Miami, FL, 33149, USA

[4]NOAA/OAR/Geophysical Fluid Dynamics Laboratory, Princeton, NJ 08540 USA

*Correspondence to*: Krysten Rutherford (krysten.rutherford@dal.ca)

**Abstract.** The global ocean's coastal areas are rapidly experiencing the effects of climate change. These regions are highly

dynamic, with relatively small-scale circulation features like shelf-break currents playing an important role. Projections can produce widely diverging estimates of future regional circulation structures. Here, we use the northwest North Atlantic, a hotspot of ocean warming, as a case study to illustrate how the uncertainty in future estimates of regional circulation manifests itself and affects projections of shelf-wide biogeochemistry. Two diverging climate model projections are considered and downscaled using a high-resolution regional model with intermediate biogeochemical complexity. The two resulting future

scenarios exhibit qualitatively different circulation structures by 2075 where along-shelf volume transport is reduced by 70% in one of them and while remaining largely unchanged in the other. The reduction in along-shelf transport creates localized areas with either amplified warming (+3°C) and salinification (+0.25 units) or increased acidification (-0.25 units) in shelf bottom waters. Our results suggest that a wide range of outcomes is possible for continental margins and suggest a need for accurate projections of small-scale circulation features like shelf-break currents in order to improve the reliability of

biogeochemical projections.

## 1 Introduction

Over the last several decades, marine ecosystem health has been significantly threatened by the extensive warming, loss of oxygen, and acidification occurring in the global ocean (Rhein et al. 2013, IPCC 2019, Tittensor et al. 2021). To anticipate the medium- and long-term impacts of these changes for mitigation and adaptation measures, reliable projections

are needed (Tittensor et al. 2021). Earth System Models (ESMs) are the backbone of such future projections, and yet these models can vary substantially in their simulated climatic outcomes (e.g., Van Coppenolle et al. 2013; Laurent et al. 2021); their differences can be attributed to, for example, differing parameterization and insufficient spatial resolution for capturing essential dynamical features of the ocean (Flato 2011).

The effects of low spatial resolution in these ESMs are particularly evident when studying the continental margins.

Continental shelf regions, which are economically important and culturally valuable, are experiencing disproportionate effects from climate change (Laruelle et al. 2010). These effects are multi-dimensional and interconnected, with linkages between

biochemical and physical changes (Bonan & Doney, 2018), but their intricacies are difficult to capture in global climate models because the models' coarse spatial resolution cannot realistically capture the dynamics of highly productive areas at the land-ocean interface (Anav et al. 2013; Bonan & Doney 2018; Holt et al. 2017; Fennel et al. 2022). It is crucial that projections are developed that accurately constrain all dynamical aspects of shelf systems along the continental margins.

While high-resolution global and basin-scale climate models are being developed (Drenkard et al. 2021; Dunne et al. 2015), the high computational cost associated with running and storing output from these models is still prohibitive for routine use. Regional models, which are computationally more affordable and allow for higher biogeochemical complexity, can accurately capture small-scale and complex circulation features and serve as a useful complement. Here, we present a case study where we use two large-domain models as forcings for a high-resolution regional model of the northwest North Atlantic, a coastal region experiencing rapid changes. The northwest North Atlantic shelf, which sustains a significant fishing economy (Brennan et al. 2016b; O'Boyle 2012), is experiencing warming at a rate exceeding the global trend (Pershing et al., 2015; Brickman et al., 2018; Alexander et al., 2020; Neto et al., 2021) and some model projections suggest the region could continue to warm at a rate three times the global average over the next century (Saba et al., 2015). Oxygen and pH are declining quickly with rates recently estimated at $-1.19 \pm 0.45$ µM $O_2$ yr$^{-1}$ (Claret et al., 2018), and between -0.1 and -0.2 units over the last century (Curran and Azetsu-Scott, 2012; Mucci et al., 2011), respectively.

The circulation in the northwest North Atlantic is characterized by the confluence of the subpolar and subtropical gyres of the North Atlantic which strongly influence the adjacent continental shelf (Figure 1, Loder et al., 1997; Hannah et al., 2001). North of Cape Hatteras the shelf is primarily impacted by the Labrador Current System (LCS), which carries cold, oxygen- and carbon-rich subpolar North Atlantic waters southwestward (Loder et al., 1998; Fratantoni and Pickart, 2007). The outer branch of the LCS traverses along the shelf break and turns southwestward at the Tail of the Grand Banks to form the shelf-break current (SC) along the Scotian Shelf; the SC separates cold and fresh shelf water from warm and salty slope waters (Beardsley and Boicourt 1981; Loder et al. 1998; Fratantoni and Pickart, 2007) and limits cross-shelf exchange (Loder et al., 2003; Hannah et al., 1996; Rutherford and Fennel, 2018). The adjacent slope water is a mixture of water masses with a clear signature of the northeastward transiting, warm, oxygen-poor Gulf Stream. The Scotian Shelf, centrally located in the northwest North Atlantic, is currently experiencing increased inflow of warm, salty slope waters leading to pronounced warming (Brickman et al., 2018; Neto et al., 2021) and deoxygenation (Gilbert et al., 2005; Sherwood et al., 2011). Some future projections indicate a significant decline in SC strength over the next century is possible, potentially accelerating warming and deoxygenation (Saba et al. 2015, Claret et al. 2018).

To highlight the role of circulation features on setting potential future states in the northwest North Atlantic, the present study focuses on downscaling results from two relatively high-resolution simulations that accurately capture the present-day strength and placement of the SC: (1) the Geophysical Fluid Dynamics Lab CM2.6 climate model's atmospheric $CO_2$ doubling experiment (GFDL CM2.6 1% yr$^{-1}$ $CO_2$ simulation; Saba et al. 2015); and (2) the Department of Fisheries and Ocean (DFO) high-resolution North Atlantic Model (Brickman et al., 2016). These physical model scenarios (hereafter simply referred to as DFO and GFDL) are used as forcings for the same regional biogeochemical model (Atlantic Canada Model,

ACM; Figure 1) for mid-century time slices (~2075). This regional model has been shown to better capture the placement and strength of the Labrador Current (Rutherford & Fennel, 2018 vs. Bourgeois et al., 2016) and more accurately represents the biological properties than global models used by the IPCC (Laurent et al., 2021). To further focus on the role of circulation features in these potential future states, the same biological future changes (from a GFDL CM2.6 1% $yr^{-1}$ $CO_2$ simulation) are applied in both scenarios. The resulting two future scenarios from the DFO and GFDL forcings (henceforth referred to as ACM-DFO and ACM-GFDL, respectively) are compared to the present-day ACM conditions. The approach of comparing multiple future scenarios within the same high-resolution biogeochemical model framework is useful for bracketing the uncertainty range of future projections and applicable to other shelf regions.

The aim of this paper is to elucidate any differences in the circulation structure between these two scenarios after downscaling and to evaluate how these differences impact shelf-wide temperature, salinity and carbonate chemistry. Differences between the ACM-DFO and ACM-GFDL simulations in this case study emphasize the ambiguity in future projections for ocean margins and illustrate how uncertain future changes to coastal circulation features may impact shelf-wide biogeochemistry. The comparison of these contrasting outcomes for the northwest North Atlantic exemplifies the essential role of circulation features that often are not resolved in global climate models and epitomizes the value of using high spatial resolution achieved by combining global model output with regional models.

## 2 Methods

### 2.1 Regional Model Description

We employ a regional biogeochemical model of the northwest North Atlantic – the Atlantic Canada Model (ACM) – which has 30 vertical levels, ~10 km horizontal resolution, and is based on ROMS v3.5, a terrain-following, free-surface, primitive equations ocean model (Haidvogel et al., 2008). The model domain includes the Gulf of Maine, Scotian Shelf, Grand Banks and East Newfoundland Shelf (Figure 1). Brennan et al. (2016a) and Rutherford and Fennel (2018) describe the physical model setup and validation, and have shown that the model represents present-day circulation patterns reasonably well. The biogeochemical model is of medium complexity, including two phytoplankton and two zooplankton groups, is fully described in Laurent et al. (2021) and based on the model described in Fennel et al. (2006, 2008). Descriptions and validation of the biological and inorganic carbon component are given in Laurent et al. (2021) and Rutherford et al. (2021), who show that the model represents the present-day biological seasonality well. The present-day model simulation was run for 16 years from 1999 to 2014, where the first year is considered model spin-up. We focus on model years 2006 to 2014 in the present study, as in Rutherford et al. (2021).

### 2.2 Downscaling for large-domain models to the regional domain

Future scenarios of two large-domain models, the Geophysical Fluid Dynamics Lab (GFDL) global climate model CM2.6 (Delworth et al., 2012; Winton et al., 2014; Dufour et al., 2015) and the North Atlantic model from Fisheries and

Oceans Canada (DFO) described by Brickman et al. (2016), were downscaled to our regional model. The regional model was run for two 16-year future time slices, representing mid-century conditions (~2075), forced by the two large-domain models. The two regional model simulations were initialized in 2065 by adding deltas from the larger-domain models (2065 minus 1999 conditions) to the 1999 regional model distributions for temperature (T), salinity (S), horizontal momentum (U, V), sea-surface height (SSH), dissolved inorganic carbon (DIC), nitrate ($NO_3$) and oxygen ($O_2$). The last 8 years of each future time slice are analyzed, allowing for 8 years of spin up, which we found to be sufficient for the dynamical adjustment of the regional model. Daily 3D model output of biogeochemical properties, dye tracers, salinity, and temperature was saved. Details about these downscaling methods are given below.

### 2.2.1 Downscaling GFDL CM2.6 climate model

The GFDL CM2.6 climate model is a coupled atmosphere-ocean-ice model, which is fully described in Delworth et al. (2012), Winton et al. (2014) and Dufour et al. (2015), and whose ocean component (MOM5; Griffies, 2012) has $1/10^o$ resolution and 50 vertical levels. GFDL performed 2 different simulations: (1) a pre-industrial control scenario, which is an 80-year simulation with atmospheric $CO_2$ held constant at pre-industrial concentrations; and (2) a warming scenario with $CO_2$ doubling where $CO_2$ is increased at an annual rate of 1% until it is doubled (model year 70), at which point atmospheric $CO_2$ is held constant for an additional 10 years.

From the GFDL warming scenario, monthly output of all physical variables (T, S, U, V, SSH) and atmospheric forcing (air temperature, air pressure, rain, radiation, wind, humidity) were interpolated to the regional model grid using objective analysis. After interpolation, the mean annual cycle was calculated over the 80-year simulation at each grid cell for both the oceanic and atmospheric variables and removed, leaving de-seasonalized gridded data. The time dimension of this de-seasonalized data was then stretched so that the doubling trajectory of atmospheric $CO_2$ closely resembles that of the RCP6.0 scenario (following Claret et al., 2018). This results in CM2.6 time being stretched by a factor of 1.903 ($t_{rcp6} = 1.903 t_{cm26} + 1947.5$) to equal RCP6.0 time.

The initial file for the time slice was created by first calculating the difference between 2065 and 1999 for each of the physical variables from the de-seasonalized monthly means and temporally stretched gridded data. This difference was then added to each of the physical variables in the 1999 regional model initial file, and the model was run for 16 years starting in 2065. The time-dependent surface and lateral boundary conditions were also taken from the de-seasonalized and temporally stretched data from CM2.6. For this, timeseries of both atmospheric and ocean variables from CM2.6 were normalized to calendar year 1999 by subtracting the 1999 de-seasonalized annual mean from the entire CM2.6 de-seasonalized timeseries for RCP6.0 years 2065-2080. These normalized timeseries were then added to the present-day climatology: for the atmospheric forcing, 3-hourly surface forcing from the European Centre for Medium-Range Weather Forecasts (ECMWF) ERA-Interim global atmospheric reanalysis data (Dee et al., 2011) from 1999-2009 was used as the baseline; for the lateral boundaries, a long-term monthly mean from the Urrego-Blanco and Sheng (2012) regional ocean model was used as the baseline climatology.

## 2.2.2 Downscaling DFO ocean model


The DFO ocean model is a 1/12-deg model of the North Atlantic Ocean on a domain covering 7-75°N latitude and 100W-25°E longitude. It has 50 vertical levels, and partial cells for the bottom layer. Forcings for the DFO ocean model's future climate simulations were derived as anomalies from the mean of an ensemble of six CMIP5 Earth System Model (ESM) future climate runs for two future periods (2046-2065 and 2066-2085), and under RCP8.5 (van Vuuren et al., 2011). The

present ocean climate was simulated using the (repeat cycle) Co-ordinated Ocean–Ice Reference Experiments (CORE) Normal Year atmospheric forcing (Large and Yeager, 2004). Future climate anomalies were added to the present climate forcing to create four future climate forcings. The resulting ocean model simulations produced climatologies for the future periods 2046-2065 and 2066-2085 for RCPs 8.5.

The difference between the DFO ocean model's RCP8.5 2066-2085 period and present-day climatology were

provided for all physical variables (T, S, SSH, U, V) as profiles at the lateral boundaries of the ACM model domain and for surface forcings (air temperature and precipitation). Other atmospheric forcings (e.g. winds, humidity, radiation) were not available; changes to these variables under the future scenario were thus assumed to be negligible. These differences were interpolated to the regional model grid and added to the present day (1999 -2014) lateral boundary conditions and surface forcings to create time-dependent future forcings. The boundary point profiles were averaged to get one profile of differences

between 2066-2085 and present-day. This average profile was added to the entire 1999 initial condition to get the future scenario initial file. The model was run for 16 years, simulating the average conditions between 2066-2085 in the larger-domain model, which is a similar period to the GFDL model time slice.

## 2.2.3 Downscaling biogeochemical conditions

The same biological initial and boundary conditions were applied to both future scenarios. These biological conditions

were taken from the GFDL CM2.6 biogeochemical model (miniBLING, see Galbraith et al., 2015; Dufour et al., 2015) output which is only available for the last 20 years of both the control and $CO_2$ doubling experiment (model years 60-80, which is approximately equal to RCP6.0 years 2060-2100) as 3-D annual means. The variables DIC, $O_2$ and $PO_4$, the latter of which was converted to $NO_3$ using the Redfield Ratio, were interpolated to the regional model grid through objective analysis. The difference between the $CO_2$ doubling experiment and the control simulation over model years 61-70 (equal to years 2065-2080

in RCP6.0 years) was calculated at every grid cell and averaged throughout that time period to get one biological delta for every grid cell, essentially estimating the difference between the 2065-2080 period and 1999. The average difference over this period was added to the 1999 initial file and boundary conditions in both scenarios. This approach assumes that the bulk of the biological changes between the control and $CO_2$ doubling experiment occur after 1999 and could therefore slightly overestimate any biogeochemical changes. In both simulations, atmospheric $p$$CO_2$ was set to follow RCP6.0 conditions with

the present-day seasonal cycle imposed (see Supporting Information in Rutherford et al., 2021, for the present-day seasonal cycle).

**2.3 Dye tracer implementation**

We additionally implemented passive dye tracers in each of the time slices, the setup of which is described in detail in Rutherford and Fennel (2018) and the dye tracer source regions are shown in Figure 1. As in Rutherford and Fennel (2018),
two types of dye tracer simulations were performed: (1) dye tracers were initialized once after model spin-up and allowed to advect and diffuse throughout the model domain, and used to visualize and quantify changes to dye tracer pathways; and (2) dye tracers were constantly reinitialized in their source region as a constant supply of the dye tracers, used to calculate dye tracer mass fractions on each of the shelves of interest (Grand Banks, Scotian Shelf and Gulf of Maine).

**3 Results**

**3.1 Projected changes to along-shelf transport and water-mass composition**

Simulation of passive dye tracers, as in Rutherford & Fennel (2018), allows visualization and quantification of circulation changes in the simulated time slices. Changes in the SC and its southwestward transport along the shelf are illustrated by the distribution of Labrador Sea (LS) dye (shown 9 months after dye initialization in Figure 2). At present (~2010; Figure 2a), the SC is intact, following the shelf edge from the Grand Banks (GB) southwestward along the Scotian Shelf. In
both future scenarios, the amount of LS dye moving along the shelf break declines. In ACM-DFO, there is a ~33% decrease in the northeastern portion of the shelf break (Figure 2c). In ACM-GFDL, LS dye disappears entirely along the Scotian Shelf, at the Tail of the Grand Banks, and in the Laurentian Channel (Figure 2e). These results are consistent with the large decline (~70%) in the southwestward volume transport along the Scotian Shelf break in ACM-GFDL versus negligible changes to the transport in ACM-DFO (see Supplement, Figure S1).

With less LS water moving southwestward in both scenarios, the presence of deep slope water (Slp-D; Slp initialization region below 200m) increases on the Scotian Shelf, particularly along the Halifax (HAL) transect in the deep basins, and along the shelf break (Figure S2). This is more pronounced in ACM-GFDL, where a larger fraction of LS water is replaced by Slp-D water. This replacement is also reflected in the dye tracer mass fractions, which provide more comprehensive information about the changes in composition of shelf waters (Figure 3). In ACM-DFO, the composition of water on the shelf
is similar to the present-day composition with only modest changes in the ratio of Slp (Slp-S+Slp-D) water to LS and Eastern Newfoundland Shelf (ENS) water throughout the shelf region and slight increases in the fraction of Slp water on the Grand Banks and the Scotian Shelf. In ACM-GFDL, the ratio of Slp water to LS+ENS water increases markedly on both the Scotian Shelf and in the Gulf of Maine, with less Slp water increases on the Grand Banks (Figure 3).

Resulting changes in temperature and salinity on the shelf in both future scenarios are summarized in Table 1. At the
surface, temperature changes are similar in both scenarios, although ACM-DFO is slightly warmer throughout the shelf. Surface salinity changes are similar on the Scotian Shelf between the two scenarios; the magnitude of surface salinity changes is however larger on the Grand Banks and in the Gulf of Maine in ACM-GFDL. The differences between the two scenarios

are most obvious in bottom waters (Table 1, Figure 4). There is increased warming throughout the shelf bottom waters in ACM-GFDL (on average +1-3ºC), with particularly large warming in the southern portion of the Scotian Shelf (+2.8ºC on

average with maxima of up to 5ºC). These changes are also reflected in salinity. The southern portion of the Scotian Shelf, the Gulf of Maine and the Laurentian Channel show pronounced salinification (by approximately +0.2-0.3) in ACM-GFDL whereas the Grand Banks and the northern portion of the Scotian Shelf are considerably fresher (by about -0.1-0.2). These large differences in bottom water properties between the two scenarios are a result of diminished transport of cool, fresh water southwestward along the shelf break in ACM-GFDL, leading to warmer, saltier water on the southern portion of the shelf and

in the Gulf of Maine, particularly in the deep basins (see LS vs Slp-D dye; Figure S2).  The resulting future changes in temperature and salinity are more fully described in the Supporting Information (see Figure S3).

**Table 1: Spatially and temporally averaged changes to temperature and salinity in each of Grand Banks (GB), Scotian Shelf (SS) and Gulf of Maine (GoM).**

|  |  | Average change throughout water column | | | Average change in surface waters | | | Average change in bottom waters | | |
|---|---|---|---|---|---|---|---|---|---|---|
|  |  | GB | SS | GoM | GB | SS | GoM | GB | SS | GoM |
| ACM-DFO | Temperature (ºC) | +1 | +1 | +1 | +2.1 | +2.7 | +1.8 | +0.1 | +0.2 | +0.4 |
|  | Salinity | +0.05 | -0.05 | - 0.05-0.1 | -0.11 | -0.15 | 0.13 | +0.02 | +0.01 | +0.02 |
| ACM-GFDL | Temperature (ºC) | +1.5 | +2.5 | +2 | +1.3 | +2.2 | +1.6 | +1.3 | +2.8 | +2.2 |
|  | Salinity | -0.45 | -0.05 | +0.25 | -0.58 | -0.19 | +0.27 | -0.20 | +0.20 | +0.28 |


### 3.2 Projected changes to carbon properties

Since differences between the two future scenarios in temperature and salinity are larger in bottom waters, we focus most of our remaining analysis on comparisons of bottom water properties on the shelves. In contrast to temperature and

salinity, where there are substantial differences between the scenarios in bottom waters, bottom DIC concentrations are relatively similar between the two scenarios with increases of 74, 56, and 61 mmol C m$^{-3}$ in ACM-DFO and 69, 59, and 65 mmol C m$^{-3}$ in ACM-GFDL for Grand Banks, Scotian Shelf, and Gulf of Maine, respectively. ACM-DFO does have higher bottom DIC than ACM-GFDL across the East Newfoundland Shelf and Grand Banks, whereas ACM-GFDL has higher bottom DIC into the Gulf of St. Lawrence, Gulf of Maine and on the Scotian Shelf (Figure 5 e).

There are more notable differences in bottom pH. In ACM-GFDL, bottom waters are more acidic, particularly on the Grand Banks and on the more northern portion of the Scotian Shelf (Figure 5f). Figure 5a, b shows the actual bottom pH values in the two future scenarios and highlights the most acidic regions, which are the tip of Grand Banks, coastal areas in the Gulf of St. Lawrence, Gulf of Maine, and on the Scotian Shelf, and the more northern portion of the Scotian Shelf. In general, these regions are more acidic in ACM-GFDL, reaching minimum pH values of ~7.7 pH units (e.g., GB, northeastern Scotian Shelf) compared to ~7.8 pH units in ACM-DFO. The lowest pH values are in the Gulf of St. Lawrence (7.6 in the ACM-GFDL scenario; 7.7 in ACM-DFO). It is important to note that the regions in ACM-GFDL with more acidic waters than in ACM-DFO do not strictly align with waters higher in DIC than ACM-DFO (Figure 5e, f). Like bottom salinity and temperature, bottom pH in ACM-DFO is more uniform than in the ACM-GFDL, where a stronger north to south gradient in pH (more acidic to less acidic) is present (Grand Banks and northeastern Scotian Shelf vs. southwestern Scotian Shelf and Gulf of Maine).

Under both future scenarios the whole shelf acts as large net source of $CO_2$ (Table S3) following large increases in DIC and temperature throughout the region. ACM-GFDL has larger increases in air-sea $CO_2$ flux than ACM-DFO, most notably on the Grand Banks, and to a lesser extent in the Gulf of Maine. The changes in air-sea $CO_2$ flux on the Scotian Shelf are similar in both scenarios. There is also a gradient from stronger net outgassing on the Grand Banks to weaker net outgassing in the Gulf of Maine in both scenarios (Table S3). Under present-day conditions, the model estimates only the Scotian Shelf is a large net source of $CO_2$, while the Gulf of Maine and Grand Banks are estimated to act as net sinks (Table S3; see also Rutherford et al., 2021; Rutherford and Fennel, 2022). Overall, differences in the air-sea $CO_2$ flux between the two scenarios are relatively small compared to the total increase in surface air-sea $CO_2$ fluxes from present-day.

**3.3 Effects of altered water-mass composition**

The 70% decline in southwestward volume transport along the Scotian Shelf in ACM-GFDL (Figure S1) consequently results in changes to the water-mass composition on the shelf, as previously illustrated in Figure 3 and further summarized in Figure 6. Figure 6 illustrates how to interpret the changes in dye tracer mass fractions as it relates to the dominant end-members in the region: subpolar North Atlantic water (ENS,LS) and warm, salty slope water (Slp). With similar southwestward volume transport in ACM present-day and ACM-DFO, the water-mass composition and transit pathways are similar (Figure 6a,b). Conversely, in ACM-GFDL with a large decline in southwestward transport of subpolar North Atlantic water, there is a large decline in both ENS and LS dye and an increase in Slp dye reaching the Scotian Shelf and Gulf of Maine. These changes result in an altered water-mass composition on the shelf system as a whole, but particularly on the Scotian Shelf and Gulf of Maine (Figure 6c).

Differences in temperature, salinity and pH between these simulations are most obvious in bottom waters which are less influenced by atmospheric inputs; these differences are summarized in Figure 7. Both present-day and ACM-DFO simulations have similar bottom temperature and salinity spatial trends (Figure 7a). Temperature is coolest on the more northern part of the shelf system (Grand Banks, northern Scotian Shelf ($SS_{north}$)) and warmest on the most southern part of the shelf system (Gulf of Maine, southern Scotian Shelf ($SS_{south}$)). There is less spatial variability in salinity, but SSnorth is the

freshest area due to the large influence from the Gulf of St. Lawrence. $SS_{south}$ is about 0.5 salinity units saltier than SSnorth. In ACM-GFDL, there are larger differences in both bottom water temperature and salinity (Figure 7a). Although the same

north-south trend in bottom temperature is present in ACM-GFDL, the southern shelves (SS, GoM) are over 2°C warmer than at present-day and ACM-DFO. This is in contrast to surface waters where ACM-DFO is warmer throughout the shelf system than ACM-GFDL (Table 1). There are additionally large changes in bottom salinity in ACM-GFDL. While the Grand Banks become slightly fresher and the northern Scotian Shelf is relatively unchanged, the southern Scotian Shelf and Gulf of Maine both become saltier by nearly 0.5 and 0.3 units, respectively. As a result, $SS_{south}$ is nearly 1 unit saltier and ~3°C warmer than

$SS_{north}$ in ACM-GFDL versus 0.5 units saltier and 2°C warmer in ACM-DFO and at present-day. The changes in temperature and salinity in bottom waters in ACM-GFDL create a larger difference between $SS_{north}$ and $SS_{south}$ than in the present-day simulation and ACM-DFO. This change in spatial variability is reflected in changes in bottom pH (Figure 7b).

Figure 8 further illustrates these spatial trends as they relate to changes in water-mass composition (i.e. changes to the ratio of LS+ENS to Slp dye). Values of LS+ENS:Slp less than one indicate areas that have become dominated by warm,

salty slope water; conversely, areas with values greater than one are dominated by subpolar North Atlantic water. Only in ACM-GFDL are areas (GoM, $SS_{south}$ and SS as a whole) more dominated by Slp waters. In both ACM-GFDL and ACM-DFO, all shelf areas shift towards lower LS+ENS:Slp values; however, this shift is much larger in ACM-GFDL. Larger dominance of slope water tends to correspond to warmer bottom waters (Figure 8a) throughout all simulations. Although there is less of a clear trend across all simulations in salinity (Figure 8b), regions with LS+ENS:Slp values less than one have the largest

bottom water salinities. In terms of biogeochemistry, bottom DIC is relatively uniform across different water-mass compositions, and any differences in bottom DIC between the two future scenarios are small in comparison to overall increases in DIC in both ACM-DFO and ACM-GFDL from present-day (Figure 8c). Both ACM-DFO and ACM-GFDL have similar overall declines in pH throughout the system (Figure 8d), likely reflective of similar increases in bottom DIC. However, there is larger variability in bottom pH in ACM-GFDL that follows the variability of temperature and salinity associated with larger

proportions of slope water

## 4 Discussion

The two future downscaled scenarios presented here project diverging estimates of the future regional circulation structure. In ACM-DFO, changes to the southwestward volume transport are modest (Figure S1) and the delivery of LS water to the Scotian Shelf is reduced by about 30% compared to the present (Figure 2). Conversely, ACM-GFDL exhibits a 70%

decline in volume transport along the Scotian Shelf (Figure S1) and LS water practically disappears along the break of the Scotian Shelf (Figure 2). Instead, slope water becomes the largest contributor of all the endmembers (Figure 3).

Comparison of these two simulations shows that bottom water properties are strongly determined by the circulation, because the permanent density stratification on the shelf insulates bottom waters from atmospheric influences. In ACM-DFO, where the shelf-break current is intact, temperature and salinity changes are small in the shelf bottom waters (Figure 4).

Conversely, in ACM-GFDL with the shelf-break current nearly vanishing, there is extensive bottom water warming on the shelves, in some locations by up to +5°C. Although one could argue that these larger increases in bottom water temperatures in ACM-GFDL could be due to atmospheric inputs, ACM-DFO actually has larger surface water warming than ACM-GFDL (Table 1). It is thus more likely that these large increases in bottom temperature are a result of higher proportions of slope water on the shelves, which is a warmer and saltier end-member (Figure S3). Slp-S and Slp-D end-members did warm slightly
more in ACM-GFDL than in ACM-DFO, which is likely also contributing to bottom waters in ACM-GFDL being warmer across the shelf system. Perhaps even more interesting are the changes in bottom water salinity, since salinity acts as a more conservative tracer of water masses than temperature. In the Gulf of St. Lawrence, the southern portion of the Scotian Shelf and in the Gulf of Maine, there are large salinity increases (up to +0.28) under ACM-GFDL. The regions experiencing higher projected salinification (Figure 4 and Table 1) also have larger amounts of Slp-D water that has replaced LS water (Figure 3).
Surface temperature and annual air-sea $CO_2$ flux differ less between the two future climate scenarios (Table 1 and Table S1) indicating that the shelf-break current strength is less of a control on the surface carbon budget.

These differences in the shelf-break current strength have large impacts on the spatial variability of the bottom water pH. In ACM-GFDL, the weak shelf-break current promotes significant inflow of slope water, largely impacting the southwestern portion of the Scotian Shelf and the Gulf of Maine (Figure 4). This increased inflow of warm, salty slope water
amplifies the presently existing disparity between the southwestern and northeastern Scotian Shelf in terms of temperature and salinity (Figures 7, 8). With a weakened shelf-break current, the southwestern portion of the Scotian Shelf behaves more similarly to the Gulf of Maine, and the northeastern portion remains more similar to Grand Banks with additional influence from the Gulf of St. Lawrence . This north-south trend is also evident in bottom water pH (Figures 5 and 7). Although the overall decline in pH is strongly dependent on increased DIC throughout the model domain and the magnitude of this decline
is similar in both ACM-DFO and ACM-GFDL, the weakened shelf-break current in ACM-GFDL creates localized regions where increased inflow of warm, salty slope water thermodynamically dampens the acidification seen throughout the rest of the shelf system, compared to more uniform changes to pH in ACM-DFO. As a result, in the future scenario with a weakened shelf-break current the shelf regions are divided into either regions that are warmer, saltier, and less acidic (southwestern shelves) or regions that are cooler, fresher, and more acidic (northeastern shelves). The regions that are more highly affected
by warm, salty slope water (central Gulf of Maine, southwestern Scotian Shelf) tend to experience higher warming but less acidification than regions with larger subpolar North Atlantic water influence.

The resulting differences between these two future scenarios could create widely diverging futures for shelf-wide ecology. Under a scenario with less southwestward transport and increased inflow of warm, salty slope water, there would likely be more significant changes to the shelf ecosystem as a whole. For example, the copepod population on the northeastern
Scotian Shelf (i.e., *Calanus glacialis*, *Calanus hyperboreus*) has historically been set with the delivery of cold water from the Gulf of St. Lawrence and Labrador Current (Tremblay and Roff, 1983; Sameoto and Herman, 1990; Herman et al., 1991; Sameoto and Herman, 1992). With less southwestward transport, the delivery of these species could significantly decline. Furthermore, *Calanus finmarchicus* copepods, which are largely found on the southwestern Scotian Shelf and into the Gulf of

Maine, could see impacts to their population as a result of increased warm and salty slope water infiltrating these areas. In fact, since 1999, declines have already been observed in the *Calanus finmarchicus* populations, and they have been replaced by smaller and warmer-water copepods, like *Pseudocalanus* spp. (Lotze et al. 2022; Bernier et al. 2018; Pershing et al. 2021). These shifts in the foundation of the food-web will have cascading effects on higher-up consumers like forage fish, larval Atlantic cod and North Atlantic right whales (Lotze et al. 2022; Pershing et al. 2021); these shifts could be significantly amplified under a future scenario with large changes to the circulation regime.

The effects of a weaker shelf-break current are also seen in the creation of localized areas experiencing either higher bottom warming or increased acidification, which will additionally be important for management and adaptation measures. Identifying these areas would be difficult to elucidate with coarse-resolution global climate models alone. Already at present day, it has been found that the southwestern and northeastern Scotian Shelf can be delineated ecologically across multiple species, divided into northern and southern subpopulations (Frank et al. 2006; Stanley et al. 2018). From our results, we might anticipate large differences between how the southern and northern subpopulations adapt under a future scenario with a weaker shelf-break current. For example, benthic calcifying species, such as adult American lobster and European green crab, in the northern subpopulation might experience larger habitat shifts than those same species in the southern subpopulation due to more acidification on the northern shelves. Meanwhile, surface dwelling species will be most impacted by atmospheric inputs regardless of location and circulation regime. It is important to note that these are not the only stressors that will be impacting localized areas on the shelf here. Deoxygenation, for example, has already been observed in regions experiencing larger influence from warm, salty slope water, such as into the Laurentian Channel and the deep basins along the southwestern Scotian Shelf (e.g. Gilbert et al. 2005, Brennan et al. 2016b, Claret et al. 2018) and oxygen is anticipated to continue to decline under a future scenario with less southwestward volume transport (Claret et al. 2018).

The validity of each of these future scenarios for the northwest North Atlantic is of course difficult to assess. The future time slices presented here are relatively short, and previous studies have shown that natural climate variability can dominate climate signals in simulations of shorter timeframes, such as in the present study (e.g. Drenkard et al. 2021; Deser et al. 2012a,b). The results presented here would therefore include natural climate signals, and potentially over- or underestimate future changes (Drenkard et al. 2021). These two cases should therefore be viewed as potential bookends for a wide range of possible outcomes for the region. The direct comparison of these two potential fates in the same high-resolution regional model is an effective approach that allows comparison of the implications of different future projections for coastal areas.

The findings from this case study, although region-specific, hold relevance for the global scale. Continental margins are already rapidly and significantly experiencing the impacts of climate change, and the northwest North Atlantic shelf is just one example. These regions are highly dynamic, often with relatively small-scale circulation features playing an important role, and this case study highlights the diverging estimates of these essential coastal circulation features under future climate scenarios. Studies such as this can help us better understand the range of possible outcomes for these important coastal regions and start to delineate what factors will be dominantly controlling different habitats and species. Our results further emphasize

the need to better constrain projections of small-scale circulation futures, such as shelf-break currents, which will overall help to decrease the uncertainty of biogeochemical projections for shelf regions.

**Author Contributions.** KF conceptualized the project. KR carried out the model simulations and analyses. LGS helped create model forcing files. JJ contributed GFDL model output. KR and KF discussed results and wrote the paper. All co-authors contributed to editing the manuscript.

**Competing Interests.** The contact author has declared that neither they nor their co-authors have any competing interests.

**Code availability.** The ROMS model code can be accessed at https://www.myroms.org (Haidvogel et al., 2008). Here version 3.5 was used.

### Acknowledgements

Charles Stock provided helpful comments and suggestions on an earlier version of the manuscript. Dave Brickman is
acknowledged for providing the model output used for the ACM-DFO scenario, and for providing comments on an earlier version of the manuscript. We would like to acknowledge the use of scientific colourmaps roma, lajolla, batlow, vik and lapaz (Crameri, 2018) used in this study. We are additionally grateful for the three reviewers of our paper and their constructive feedback.

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

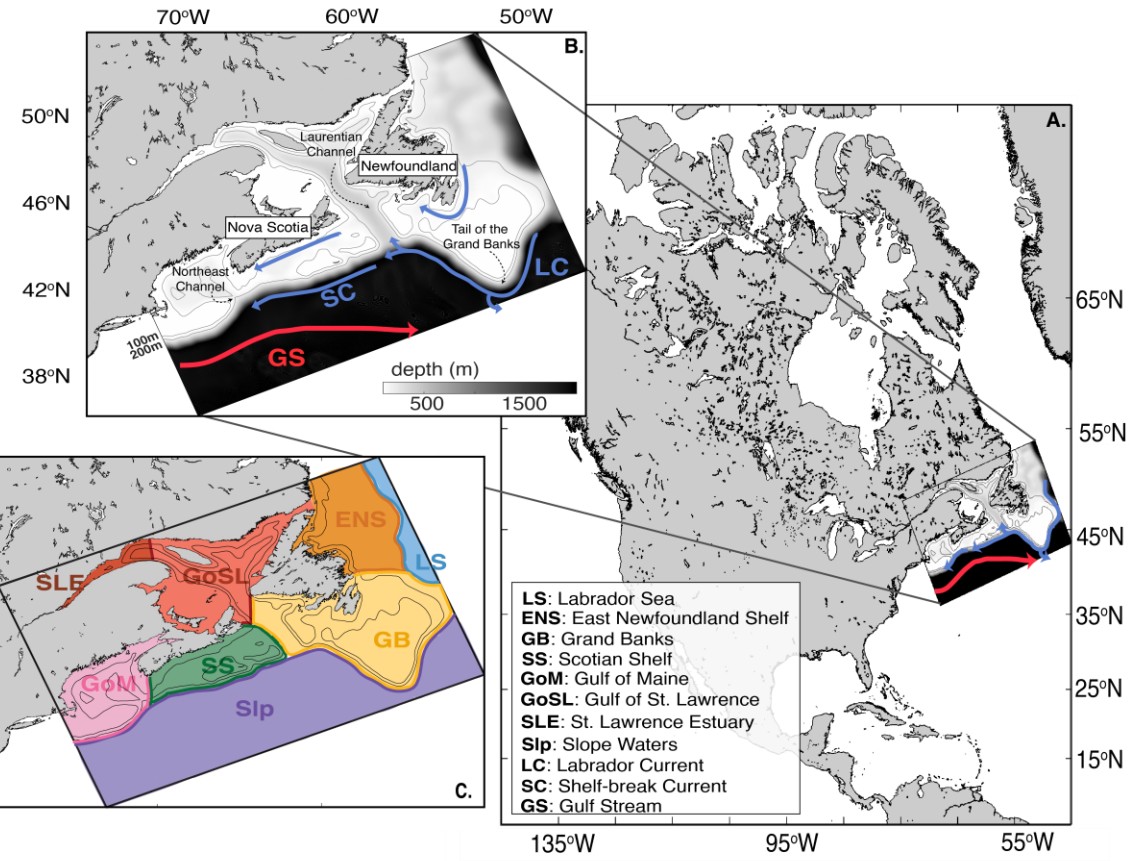

**Figure 1: (A) Map of North America with the location of the regional model indicated. (B) Regional model domain with key circulation and geographical features. (C) Dye tracer initialization regions. Slp (slope waters) are further divided into two depth levels: Slp-S, 200m and above, and Slp-D, below 200m.**


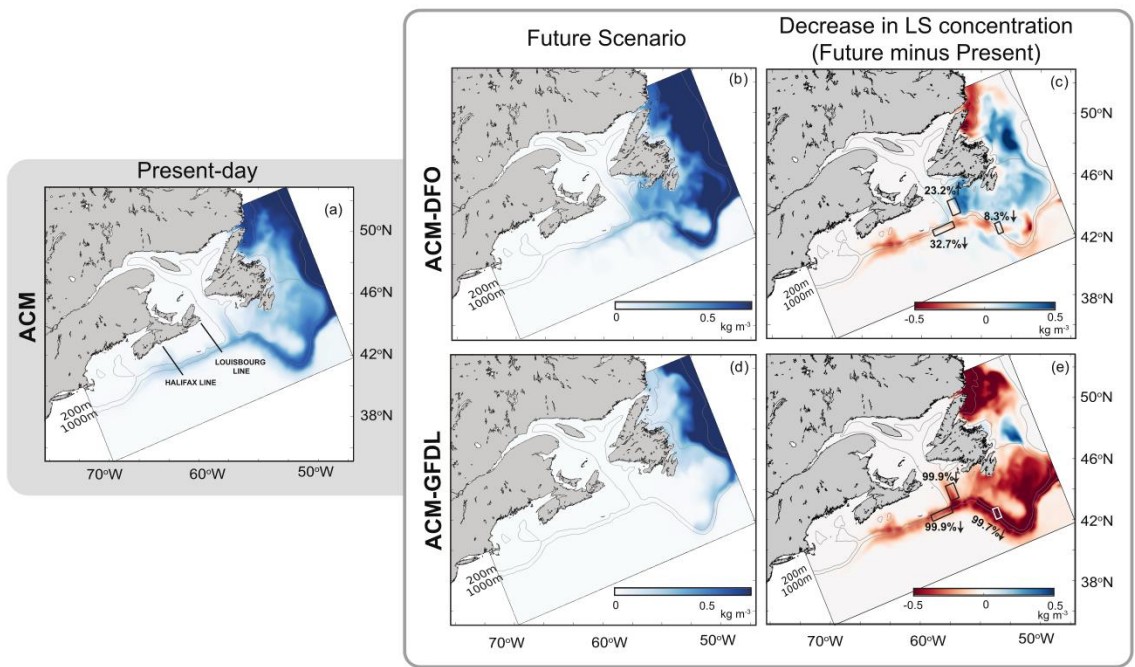

Figure 2: Time slices of vertically averaged concentration of Labrador Sea (LS) dye in (a) present day and two future (middle panels, b and d) scenarios at about 9 months after dye tracer initialization. Right panels (c and e) show the decrease in vertical mean LS dye concentration between future and present day. Three boxes indicate regions of interest with the average % decrease in water-column-averaged LS dye. Panel (a) indicates transect locations (HAL, LOU) for Figure S2.

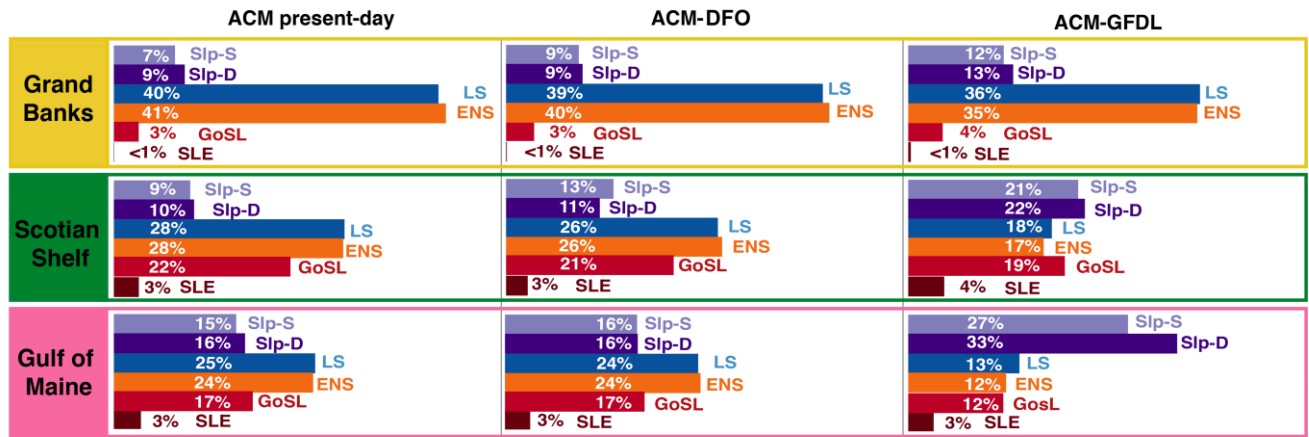

Figure 3: Mass fractions for (top to bottom) Grand Banks (GB), Scotian Shelf (SS), and Gulf of Maine (GoM) in each time slice: (left to right) ACM present day, ACM-DFO future scenario and ACM-GFDL future scenario. End-members and shelf locations are indicated in Figure 1.

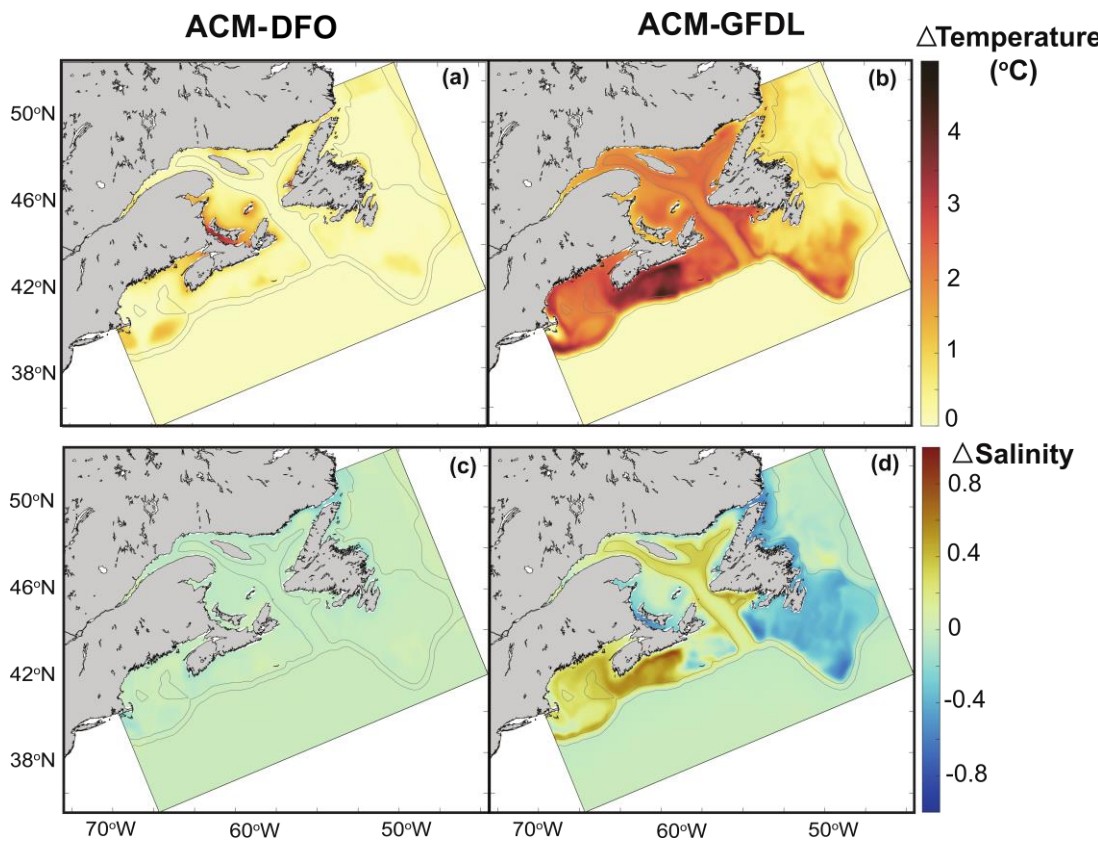

Figure 4: Bottom average changes (future minus present) in temperature (top) and salinity (bottom). Left panel is ACM-DFO and right panel is ACM-GFDL.

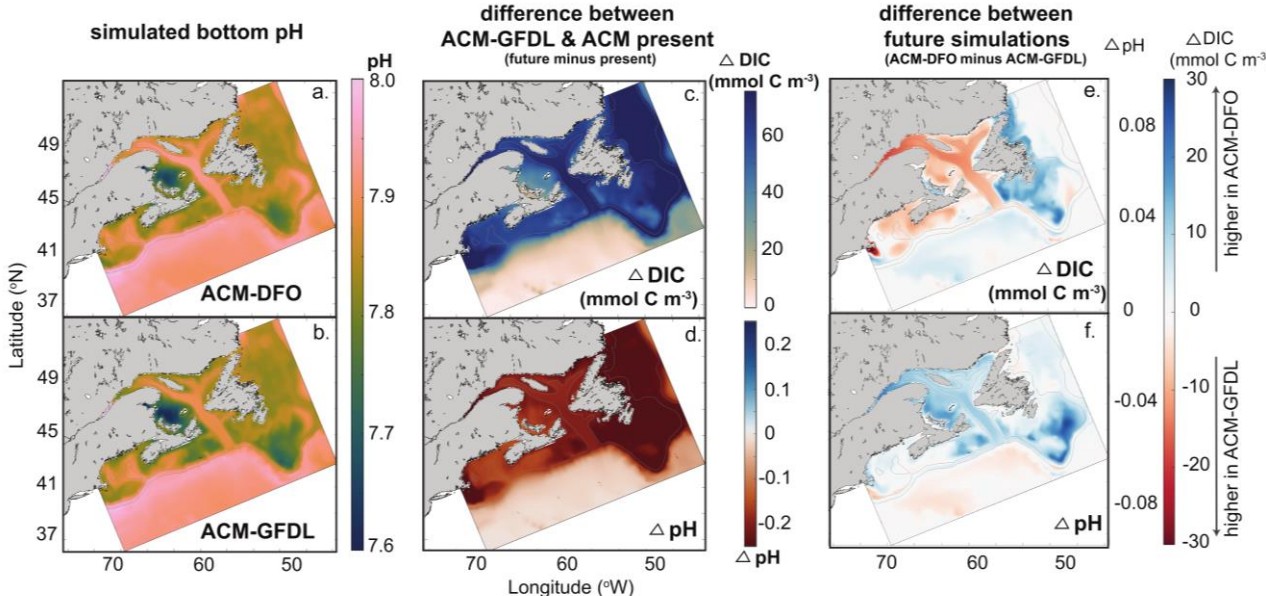

**Figure 5: Left panel: Bottom pH in two future scenarios, (a) ACM-DFO and (b) ACM-GFDL. Middle panel: Difference between ACM-GFDL and ACM present-day for bottom (c) dissolved inorganic carbon (DIC) and (d) pH. Right panel: Difference between two future scenarios (ACM-DFO minus ACM-GFDL) for (e) bottom DIC and (f) bottom pH.**

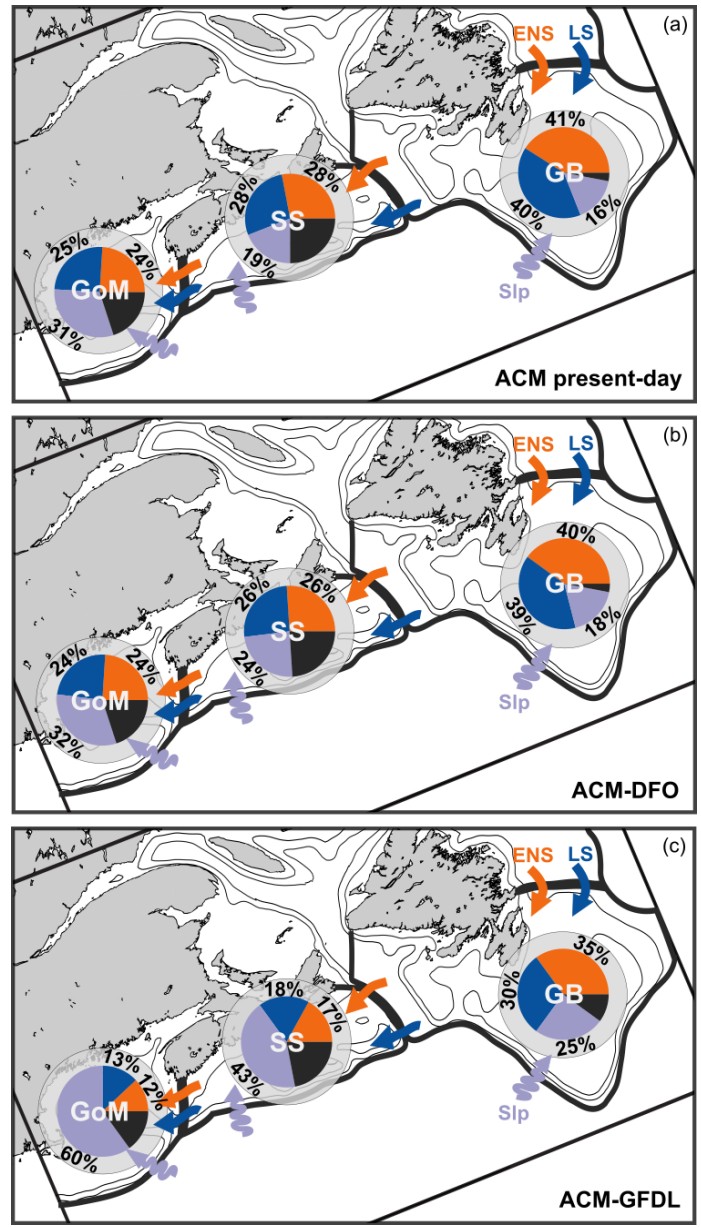

**Figure 6: Schematic representation of the water-mass composition in each simulation. Numbers represent the mass fractions described in Figure 3. Arrows are not meant to indicate exact location of water flow.**

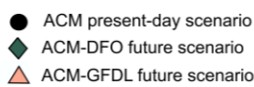

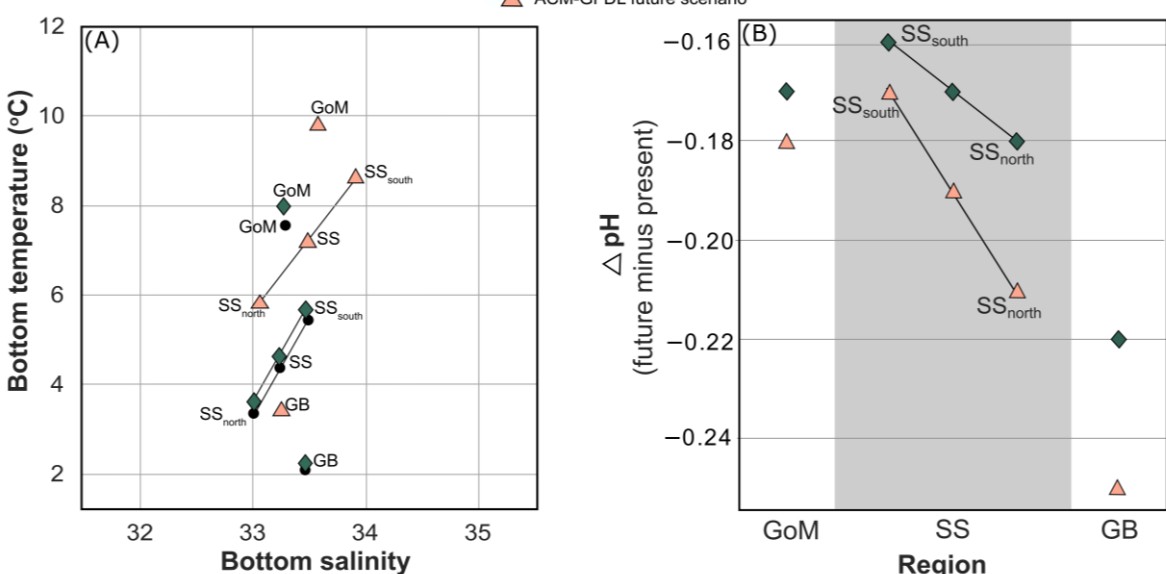

**Figure 7: (A)** Bottom temperature versus bottom salinity and **(B)** the change in bottom pH (future minus present) for Grand Banks (GB), Scotian Shelf (SS) and Gulf of Maine (GoM). The Scotian Shelf is additionally subdivided into the northern Scotian Shelf ($SS_{north}$) and southern Scotian Shelf ($SS_{south}$) in each panel to illustrate differences in spatial variability in each simulation.

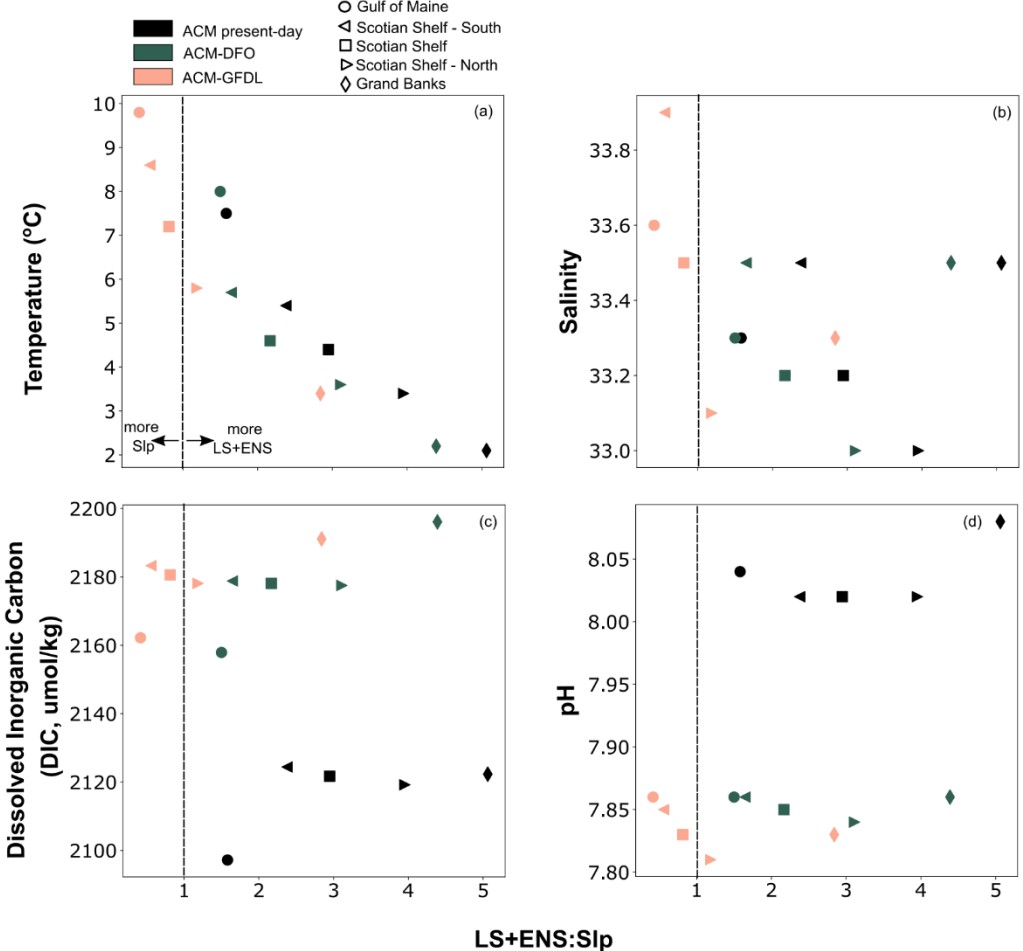

**Figure 8: Effects of different LS+ENS:Slp ratios on bottom variables - (a) temperature, (b) salinity, (c) dissolved inorganic carbon (DIC), and (d) pH - in each simulation. LS+ENS:Slp ratios above 1 indicate areas that are dominated by subpolar North Atlantic waters (LS and ENS waters); ratios below 1 indicate areas that are dominated by warm, salty slope water (Slp-S and Slp-D).**