# Peer review of "Uncertainty in the evolution of northwest North Atlantic circulation leads to diverging biogeochemical projections"

_EGUsphere, 2023_

## Author Response (AR1)

**Uncertainty in the evolution of northwest North Atlantic circulation leads to diverging biogeochemical projections**

(Reviewer's comments in black text; Responses to comments in blue text)

**Responses to Comments from Reviewer #1**

The authors downscaled two future climate projections to the Atlantic Canada domain to characterize the future states of the physical and biogeochemical environments. Their results show quite different outcomes of future climate states in the region. While the topic is interesting, the manuscript is mostly descriptive and speculative and does not provide much insightful new knowledge or sufficient explanations of the results. Because it is known that future climate projections can be widely different across models (hence we have CMIP), downscaled projections to a regional model being different should not be surprising. **The weakness of this manuscript, however, is the lack of robust connections showing that the changes of circulation (shelf-break currents) cause the different biogeochemical projections.** For example, **how does the changes in the shelf-break currents affects the temperature and salinity over the shelf**? Note shelf-break currents and along-shelf currents are different. Tracer released over the Labrador continental slope is expected to move along the slope and shelf-break, however, tracer concentrations from the two simulations doesn't explain the changes of physical and biogeochemical environment on the continental shelf, which seems to be the focus of this manuscript. A **more relevant analysis can be to compare the distribution of ENS tracer (on the shelf)**. To understand the causes of the simulated temperature and salinity changes, **budget calculations including the along-shelf and cross-shelf advective fluxes as well as air-sea fluxes are needed.** Otherwise, claiming shelf-break currents causing the changes is unsupported. Similarly, **how the diverging projections of temperature and salinity lead to diverging biogeochemical projections, e.g., PH and DIC, needs to be supported with actual analysis**.

**Response**: We thank the Reviewer for taking the time to review our manuscript. We've outlined replies to their major comments below.

Regarding connections between changes in circulation and the resulting effect on the biogeochemical projections: we added the following figures and text into a third Results subsection. We have moved Figure 6 (now referred to as Figure 7) to this subsection.

**"3.3 Effects of altered water-mass composition**

[revised manuscript text omitted]

We additionally updated Lines 222ff (lines 280ff in revised manuscript; changes in bold italics):

*"Conversely, in ACM-GFDL with the shelf-break current nearly vanishing, there is extensive bottom water warming on the shelves, in some locations by up to +5ºC. **Although one could argue that these larger increases in bottom water temperatures in ACM-GFDL could be due to atmospheric inputs, ACM-DFO actually has larger surface water warming than ACM-GFDL (Table 1). It is thus more likely that these large increases in bottom temperature are a result of higher proportions of slope water on the shelves, which is a warmer and saltier end-member (Figure S3). Slp-S and Slp-D end-members did warm slightly more in ACM-GFDL than in ACM-DFO, which is likely also contributing to bottom waters in ACM-GFDL being warmer across the shelf system.**"*

And lines 231-235 (lines 294ff in revised manuscript; changes in bold italics):

*"This increased inflow **of warm, salty slope water amplifies the presently existing** disparity between the southwestern and northeastern Scotian Shelf **in terms of temperature and salinity** (Figures 7, 8).  With a weakened shelf-break current, the southwestern portion of the Scotian Shelf behaves more similarly to the*

*Gulf of Maine, and the northeastern portion **remains more similar to Grand Banks with additional influence from the Gulf of St. Lawrence**. This north-south trend is also evident in bottom water pH (Figures 5 and 7). **Although the overall decline in pH is strongly dependent on increased DIC throughout the model domain and the magnitude of this decline is similar in both ACM-DFO and ACM-GFDL,** the weakened shelf-break current in ACM-GFDL creates localized regions **where increased inflow of warm, salty slope water thermodynamically dampens the acidification seen throughout the rest of the shelf system**, compared to more uniform changes to pH in ACM-DFO."*

Regarding the distribution of ENS tracer vs LS tracer: We chose to display distributions of the LS tracer in Figure 2 as this dye tracer highlights the shelfbreak current (or lack thereof) best. The ENS tracer is more representative of the along-shelf transport on the inner shelf, which is not the intended purpose of Figure 2. Changes in ENS dye is included in the dye tracer mass fractions shown in Figure 3, which includes the fraction of ENS and LS dye on Grand Banks, Scotian Shelf and Gulf of Maine.

Other comments:

Line 18: "*Our results illustrate that a wide range of outcomes is possible for continental margins*" This is extrapolation and unsupported.

**Response**: We updated this sentence to read: "*Our results **suggest** that a wide range of outcomes is possible for continental margins ...*"

Line 61: "*Future projections indicate a significant decline in SC strength over the next century potentially accelerating warming and deoxygenation (Saba et al. 2015, Claret et al. 2018).*" Isn't the projection of SC strength model-dependent, as mentioned in the abstract?

**Response**: We have updated this sentence to read: "***Some** future projections indicate a significant decline in SC strength…*"

Line 231: "*This localized increased inflow creates an even larger disparity between the southwestern and northeastern Scotian Shelf than what is currently present*". Localized increase inflow is not shown. This statement (and the paragraph) is unsupported.

**Response**: Increased inflow is indeed shown in Figure S2 where we show that there is a larger concentration of Slp-D dye in the deep basins along the Halifax transect in both scenarios, but particularly so in the ACM-GFDL scenario. This figure additionally shows that there is a large decrease in LS dye along the shelf break of the Scotian Shelf. Additionally, the changes outlined above in response to the Reviewer's first comments should address their concern about whether there is a disparity between SS$_{south}$ and SS$_{north}$, and how this relates to changes in water-mass composition.

**Uncertainty in the evolution of northwest North Atlantic circulation leads to diverging biogeochemical projections**

(Reviewer's comments in black text; Responses to comments in blue text)

**Responses to Comments from Reviewer #2**

The study uses a regional ocean model to investigate how the future ocean conditions in the northwest North Atlantic, like the emergence of warming and acidification, are controlled by climate-induce changes in the local circulation. The study demonstrates that a plausible increase in the slope water contribution to the Scotian Shelf associated with a weakening in the shelf break current, can drive enhance bottom water warming and salinification, and lead to localised regions of reduced/increased acidification (with less acidic regions being co-located with warmer regions).

The study will further our understanding of the response of the shelf seas and coastal regions to climate change (which currently is poorly understood). The use of somewhat "idealised" and targeted simulations (e.g., forcing the regional model projections with the same biogeochemical conditions but different physics-dynamics-circulation conditions), in my opinion, is a strength of the study and enables to inform on control mechanisms. The conclusions are well supported by the analysis and figures, and to the most part the manuscript is well written. However**, the description for the implementation of the downscaling experiment is difficult to follow (at least to me) and I am still unsure if I understood the implementation of the forcing for the projections with the regional model correctly**. Hence, I recommend the following revisions for clarity.

**Response**: We thank the Reviewer for their helpful comments. Please see our responses to each of their individual comments below.

**Specific Comments**

**1. Description of the downscaling experiments with the regional model:** I suggest that section 2.2 is reorganised, restructured and re-written for clarity. I suggest that section 2.2 is separated **into two subsections** that each separately describe the two experiments**: 2.2.1 downscaling using forcing from the GFDL-1%pCO2 increase per year for both physics and biogeochemistry; and 2.2.2 downscaling using forcing from the DFO under RCP 8.5** (to me the DFO model projections and their set-up was somewhat unclear) for the physics but using the GFDL-1%pCO2 increase per year for the biogeochemistry. **Please see specific comments below, but consider re-writing the entire section as to provide a clearer description for the set-up of your experiments.**

**Response**: Creating two subsections to describe the two experiments is an excellent suggestion, and we have implemented this.

1.1 Lines 101-103 and 117-119:  To me, it is not clear what "adding the anomaly (or delta) to the 1999 distribution or to the 1999 initial file" means and what this 1999 initial file/distribution corresponds to? Do you mean that the trend from the GFLD projection (essentially the de-seasonalised anomaly at 2065 relatively to 1999) was added to the 1999 conditions from the present day run with the regional model? Or do you mean that this trend was added to the 1999 conditions from the GFDL run itself (such as to keep a constant seasonal cycle?). **Please I suggest that you clarify what this 1999 initial file/distribution corresponds to.**

**Response**: The de-seasonalized anomaly at 2065 relative to 1999 in GFDL was added to the 1999 conditions from the present-day regional model run. We updated the text accordingly.

*Lines 101-103 (lines 99-101 in revised manuscript): "The two regional model simulations were initialized in 2065 by adding deltas **from the larger-scale models** (2065 minus 1999 conditions) to the 1999 **regional model** distributions for temperature (T), salinity (S), horizontal momentum (U, V), sea-surface height (SSH), dissolved inorganic carbon (DIC), nitrate (NO3) and oxygen (O2)."*

*Lines 117-119 (line 119ff in revised manuscript): "The initial file for the time slice was **created by first calculating the difference between 2065 and 1999 for each of the physical variables** from the de-seasonalized monthly means and temporally stretched gridded data. **This difference was then added to** each of the physical variables **in** the 1999 **regional model** initial file, and the model was run for 16 years starting in 2065."*

**1.**2 Lines 120-121, surface and lateral boundary conditions: I am confused here. If I understood correctly, for the boundary conditions you do not use the same approach of adding "deltas" as for the initial conditions? If yes why not? Also, the text implies that for the ocean boundary conditions and atmospheric forcing you use directly the de-seasonalised GFDL outputs such that the imposed atmospheric and oceanic forcing for the ACM projections does not include any seasonal cycle? I am not sure that makes sense to me, so probably I have misunderstood of how the atmospheric and oceanic forcing is imposed at the open boundaries in the future time-slices experiments. **Please, I suggest that you clarify/re-write how the atmospheric and oceanic forcing along the open boundaries is estimated and imposed in the regional model future projections. Also, it will be useful to clarify which atmospheric fields are used to force your simulations.**

**Response**: We agree, this is a bit confusing. We updated the text to explain this better (see text below). Atmospheric fields used to force the model are air temperature, air pressure, radiation, humidity, rain and wind. We do calculate the lateral and surface boundary forcing files from a similar "delta" approach, which we have hopefully explained more thoroughly below. The GFDL outputs were de-seasonalized since we reconstruct the future forcing files using the seasonality from the present-day climatology (used in the present-day ACM simulation).

*"From the GFDL warming scenario, monthly output of all physical variables (T, S, U, V, SSH) and atmospheric forcing **(air temperature, air pressure, rain, radiation, wind, humidity)** were interpolated to the regional model grid using objective analysis. After interpolation, the mean annual cycle was calculated over the 80-year simulation at each grid cell for both the oceanic and atmospheric variables and removed, leaving de-seasonalized gridded data. The time dimension of this de-seasonalized data was then stretched so that the doubling trajectory of atmospheric CO2 closely resembles that of the RCP6.0 scenario (following Claret et al., 2018). This results in CM2.6 time being stretched by a factor of 1.903 (trcp6 = 1.903tcm26 + 1947.5) to equal RCP6.0 time.*

*The initial file for the time slice was **created by first calculating the difference between 2065 and 1999 for each of the physical variables** from the de-seasonalized monthly means and temporally stretched gridded data. **This difference was then added to** each of the physical variables in the 1999 **regional model** initial file, and the model was run for 16 years starting in 2065. The time-dependent surface and lateral boundary conditions were **also** taken from the de-seasonalized and temporally stretched data from CM2.6. **For this, timeseries of both atmospheric and oceanic variables from CM2.6 were normalized to calendar year 1999 by subtracting the 1999 de-seasonalized annual mean from the entire CM2.6 de-seasonalized timeseries** for RCP6.0 years 2065-2080. **These normalized timeseries were then added to the present-day climatology: for the atmospheric forcing, 3-hourly surface forcing from the European***

*Centre for Medium-Range Weather Forecasts (ECMWF) ERA-Interim global atmospheric reanalysis data (Dee et al., 2011) from 1999-2009 was used as the baseline; for the lateral boundaries, a long-term monthly mean from the Urrego-Blanco and Sheng (2012) regional ocean model was used as the baseline climatology."*

**1.**3 Lines 124-125, DFO future projections: I am unsure what you mean by "six IPCC future climate runs", (maybe from 6 CMIP5 Earth system models?). Please, I suggest that you clarify.

**Response**: We have updated this to read "six **CMIP5 Earth System Model (ESM)** future climate runs".

**1.4** Lines 131-132: This text suggests that only the air temperature and precipitation from the DFO RCP 8.5 projection are used as surface forcing for your downscaling experiments? **What about winds, humidity, radiation? How are the other atmospheric fields/forcing imposed in the regional model?**

**Response**: Only air temperature and precipitation were available from the DFO RCP 8.5 projection, thus winds, humidity, radiation, etc. were all assumed to change negligibly in this scenario. This is of course not necessarily an accurate assumption and we added the below text to the methods to clarify.

*"Other atmospheric forcings (e.g. winds, humidity, radiation) were not available; changes to these variables under the future scenario were thus assumed to be negligible."*

**1.5** Lines 133-135: To me it is not clear why and how the conditions/fields along the lateral boundaries were averaged to get the delta added to the 1999 initial field. Are the anomalies/deltas (that are added to the 1999 initial fields) in the interior of your regional model extrapolated from the conditions along the oceanic lateral boundaries? To me that does not make so much sense and it will not lead to appropriate or consistent-to-the-forcing initial conditions for the time-sclices projections. I presume that I just have misunderstood as it is not clear and can you please re-write this part for clarity.

**Response**: David Brickman at DFO was willing to share boundary averaged deltas with us. More detailed output from his simulation is not available to us.

**2. Line 163 and Figure 2**: **Why were 9 months chosen as the timescale for which to present/discuss the averaged concentration of Labrador Sea dye after dye tracer initialization?** Is this 9-months timescale relevant in terms of the Labrador current velocities and shelf-lengthscale (i.e. travel distance) arguments? If you could please clarify.

**Response**: 9 months was chosen because we found it best illustrated the differences in the shelf-break current between the simulations. These are only snapshots of Labrador Sea dye since temporal averages of this dye tracer experiment don't necessarily make sense – the dye tracer is only initialized once and eventually leaves the model domain. We do believe, however, that these figures offer a nice qualitative complement to the more quantitative metrics like the dye tracer mass fractions and volume transport, both of which are calculated over the full simulation.

**3. Figure 2 and lines 165**: In my understanding **Figure 2c shows only the decrease in LS** concentration in the future projections, rather than the change in the future minus the present day (such that regions of increase are not shown). Comparing Figure 2a and Figure 2b it seems that they should be regions of increase in LS concentration, especially in the AMC-DFO. This can be confusing and makes it difficult to judge if the amount of LS dye moving along the shelf break declines for the AMC-DFO. **I suggest to update the figure to show the actual change (increase and decrease) rather than just the decrease.**

**Response**: We updated the figure to show both increases and decreases in LS dye concentrations, as shown below.

[Figure]

**4. Lines 227-228:** I am not sure how accurate is this statement. In my understanding, the two simulations have also very different atmospheric conditions/forcing in the future. **Are the heat, momentum and freshwater air-sea fluxes similar in the two ACM-projections? If not, I suggest to clarify that the similarity of the air-sea CO2 flux in the two ACM-projections implies that "the shelf-break current strength is less of a control for the surface carbon budget" (rather that generalise to "water properties").**

**Response**: Agree, this has been changed accordingly.

**5. Figure 5 (typo in the caption):** I believe you mean "Figure 5: Left panel … ph. Right panel….

**Response**: We thank the Reviewer for catching this!

**6. Table S3 in the supplementary Information:** For clarity, I suggest you mention in the caption that positive values indicate flux from the ocean to the atmosphere (i.e. outgassing).

**Response**: Agree, we updated the caption accordingly.

**7. Figure S3 in the supplementary information:** In the caption it is mentioned that "Open symbols indicate predicted values and filled symbols indicate actual simulated values". Can you please clarify what you mean by predicted vs simulated values here? Also, to me it seems that only filled symbols are shown in Figure S3. Additionally, I am unsure about the meaning/interpretation of the lines connecting the symbols, and of the arrows with the SLE text in Figure S3a and b. If you could please clarify what these lines and arrows represent/highlight (maybe in the caption) that would be very helpful.

**Response**: Apologies – the "open symbols" in the figure caption were a remnant from an earlier version of the manuscript that included an additional analysis that we ended up not including in the final version.

This text is now removed from the figure caption. We additionally added in text, as follows, to describe the lines and arrows, as suggested.

*Figure S3: (a) T-S and (b) T-DIC diagrams, with different symbols indicating different simulations.* **Dashed lines connect endmembers and indicate the bounds of the mixing polygon. Arrows indicate where the St. Lawrence Estuary (SLE) endmember lies outside of the figure bounds.** *Panels (c) and (d) indicate changes in temperature, salinity and dissolved inorganic carbon (DIC) between the future and present-day values (future minus present).*

**Uncertainty in the evolution of northwest North Atlantic circulation leads to diverging biogeochemical projections**

(Reviewer's comments in black text; Responses to comments in blue text)

**Responses to Comments from Reviewer #3**

The authors compared two downscaled climate model projections to evaluate the mid-century physical and biogeochemical responses in the northwest North Atlantic shelf region. They demonstrated that the two models resulted in largely different changes in along-shelf circulation that contributed to varying patterns of warming, salinification, and increased/decreased acidification.

The manuscript is well written and, to my knowledge, cites the necessary bibliography. The methods are well described and the regional model used in the manuscript is well-validated and is adequate to answer the proposed questions.

In the manuscript, the authors show that changes in along-shelf transport in the two future scenarios are not similar. While ACM-GFLD shows a nearly 70% decrease in southwestward along-shelf transport in the Scotian shelf associated with the disappearance of Labrador Sea dye in the region, ACM-DFO shows nearly no change in transport and only a 33% decrease in LS dye. The literature demonstrates that the replacement of LS water with Slope Water does impact bottom temperature, salinity, and dissolved oxygen concentration in the shelf, particularly in the channels and deep basins of the Gulf of Maine and Gulf of St. Lawrence, which partially backs the results in the study.

While this is a robust result that advances knowledge, I think that linking short timescale changes in shelf properties solely to these changes misses one step. **For example, it is not clear to me how the different changes in ocean circulation shown in the two projections are responsible for the patterns in bottom pH.** Furthermore, why are the results for **the surface properties missing in the analysis**? It seems like **surface temperature is only briefly mentioned in lines 227-228**. I believe that the missing piece that establishes the causal relationship between changes in ocean circulation and diverging biogeochemical projections could be mitigated in one of two ways: (1) the more robust calculation of fluxes and budgets on each shelf region (GoM, SSsouth, SSnorth and GB) or (2) a **more anecdotal demonstration of this relationship, perhaps following the inflow of LS water and Slope water and the consequent changes in pH and DIC**.

**Response**: We thank the Reviewer for their feedback and time to prepare a thoughtful review.

In regard to surface properties, we moved Table S2 (which summaries total, surface and bottom changes in temperature and salinity) from the supplement into the main text in Section 3.1 (now referenced as Table 1) and have added the following sentences at line 179 (line 189ff in revised manuscript):

"*Resulting changes in temperature and salinity on the shelf in both future scenarios are summarized in Table 1. At the surface, temperature changes are similar in both scenarios, although ACM-DFO is slightly warmer throughout the shelf. Surface salinity changes are similar on the Scotian Shelf between the two scenarios; the magnitude of surface salinity changes is however larger on the Grand Banks and in the Gulf of Maine in ACM-GFDL.*"

Although we already comment on surface $pCO_2$ at lines 205ff (line 225ff in revised manuscript), we additionally added the following sentence at line 190 (now line 208ff) at the start of section 3.2: "*Since*

*differences between the two future scenarios in temperature and salinity are larger in bottom waters, we focus most of our remaining analysis on comparisons of bottom water properties on the shelves.*"

In response to the comment about how the different changes in ocean circulation affect the bottom pH, we added a more anecdotal demonstration of this relationship, along the lines of the Reviewer's second suggestion. We added the following figures and text into a third Results subsection. We also moved Figure 6 (now referred to as Figure 7) to this subsection.

*"**3.3 Effects of altered water-mass composition***

[revised manuscript text omitted]

We additionally updated Lines 222ff (lines 280ff in revised manuscript; changes in bold italics):

*"Conversely, in ACM-GFDL with the shelf-break current nearly vanishing, there is extensive bottom water warming on the shelves, in some locations by up to +5ºC. **Although one could argue that these larger increases in bottom water temperatures in ACM-GFDL could be due to atmospheric inputs, ACM-DFO actually has larger surface water warming than ACM-GFDL (Table 1). It is thus more likely that these large increases in bottom temperature are a result of higher proportions of slope water on the shelves, which is a warmer and saltier end-member (Figure S3). Slp-S and Slp-D end-members did warm slightly more in ACM-GFDL than in ACM-DFO, which is likely also contributing to bottom waters in ACM-GFDL being warmer across the shelf system."***

And lines 231-235 (lines 294ff in revised manuscript; changes in bold italics):

*"This increased inflow **of warm, salty slope water amplifies the presently existing** disparity between the southwestern and northeastern Scotian Shelf **in terms of temperature and salinity** (Figures 7, 8). With a weakened shelf-break current, the southwestern portion of the Scotian Shelf behaves more similarly to the*

*Gulf of Maine, and the northeastern portion **remains more similar to Grand Banks with additional influence from the Gulf of St. Lawrence**. This north-south trend is also evident in bottom water pH (Figures 5 and 7). **Although the overall decline in pH is strongly dependent on increased DIC throughout the model domain and the magnitude of this decline is similar in both ACM-DFO and ACM-GFDL,** the weakened shelf-break current in ACM-GFDL creates localized regions **where increased inflow of warm, salty slope water thermodynamically dampens the acidification seen throughout the rest of the shelf system**, compared to more uniform changes to pH in ACM-DFO.”*

**Specific comments:**

Lines 46-50: The description of the objective of the study at this point seems redundant with the last paragraph of the Introduction. I'd suggest incorporating these sentences in the last paragraph or removing them.

**Response**: We partially removed these sentences and moved the below sentence to the end of the second last paragraph of the introduction.

“*The approach of comparing multiple future scenarios within the same high-resolution biogeochemical model framework is useful for bracketing the uncertainty range of future projections and applicable to other shelf regions.*”

Lines 149-151: Again, I do not think that it is necessary to repeat the objective of the study, especially in the Methods section.

**Response**: We removed this text.

Lines 227-228: The authors should add that surface temperature changes are not shown and air-sea CO2 flux changes are shown in Table S3.

**Response**: Surface temperature changes are shown in Table S2, which we moved to the main text (as Table 1) and added reference to here along with Table S3.

Lines 263-265: Maybe it's my lack of knowledge of ecology, but it was not clear to me why Atlantic cod and snow crab would see larger habitat shifts in the southern subpopulation in a scenario with an unaltered shelf-break current.

**Response**: We removed this text.

It seems to me like Figure 3 and Table S1 give the exact same information, so one of them can be removed (the references in the text have to be adapted accordingly).

**Response**: Yes, Figure 3 and Table S1 have the same information but displayed in a different way. We removed the table from the supplement.

Figure 5: I am curious as to why the authors chose to use blue for positive and red for negative differences (especially on panel e).

**Response**: For panel d, we believe it makes sense to use red for the negative values since negative values mean acidification (i.e., the worsening of conditions). For panels e and f, we attach no special meaning to the color choice.

Figure 5: Why didn't the authors show the difference between ACM-DFO and ACM present, as they did for ACM-GFDL in panels C and D?

**Response**: We wanted to highlight the differences between the two future simulations (ACM-DFO and ACM-GFDL; panels e and f) and we felt adding another panel would crowd the figure.

Figure 6: Why didn't the authors add the present-day pH values to panel B?

**Response**: Our intention here is to show the difference in pH between the future and present-day values. We feel that showing the differences between the two experiments and the present-day pH values best shows the differences between the two experiments. However, in our changes above, we now include pH for present-day in Figure 8.

**Technical corrections:**

Line 174: Reference to Figure 4 should be Figure 3.

**Response**: We thank the Reviewer for catching this! The figure reference has been updated.

Line 221: Reference to Figure 4 instead of Figure 5.

**Response**: Again, the figure reference has been updated.

[revised manuscript text omitted]
**  Dashed lines connect endmembers and indicate the bounds of the mixing polygon. Arrows indicate where the St. Lawrence Estuary (SLE) endmember lies outside of the figure bounds. **Panels (c) and (d) indicate changes in temperature, salinity and dissolved inorganic carbon (DIC) between the future and present-day values (future minus present).**

**Air-sea CO₂ flux**

All regions (GB, SS and GoM) experience large increases in their annual air-sea $CO_2$ flux estimates (Table S3). Overall differences in the air-sea $CO_2$ flux between the two scenarios are relatively small compared to the total increase in surface air-sea $CO_2$ fluxes from present-day. The regional model does tend to slightly overestimate surface $p$CO2 at present day (Rutherford et al., 2021) and

the DIC deltas may overestimate future DIC concentrations, therefore the magnitude of outgassing reported is potentially overestimated. However, the overall finding that atmospheric forcing is the dominant control on setting future air-sea fluxes is a robust result.

**Table S13: Annual air-sea CO₂ flux for Grand Banks (GB), Scotian Shelf (SS) and Gulf of Maine (GoM) from the present-day ACM simulation, and the two future scenarios: ACM-DFO and ACM-GFDL. Positive values indicate outgassing (ocean to atmosphere); negative values indicate ingassing.**

| | Air-sea $CO_2$ Flux (mol C m$^{-2}$ yr$^{-1}$) | | |
|---|---|---|---|
| | **GB** | **SS** | **GoM** |
| **Present-day** | -1.3 ± 0.3 | + 1.7 ± 0.2 | -0.5 ± 0.2 |
| **ACM-DFO** | + 4.2 ± 0.6 | + 3.6 ± 0.4 | + 2.6 ± 0.4 |
| **ACM-GFDL** | + 5.4 ± 0.2 | + 3.8 ± 0.2 | + 3.1 ± 0.2 |

[Figure]

555

**Figure 8: Effects of different LS+ENS:Slp ratios on bottom variables - (a) temperature, (b) salinity, (c) dissolved inorganic carbon (DIC), and (d) pH - in each simulation. LS+ENS:Slp ratios above 1 indicate areas that are dominated by subpolar North Atlantic waters (LS and ENS waters); ratios below 1 indicate areas that are dominated by warm, salty slope water (Slp-S and Slp-D).**